# Effect of Ammonia-Oxidizing Bacterial Strains That Coexist in Rhizosphere Soil on Italian Ryegrass Regrowth

**DOI:** 10.3390/microorganisms10112122

**Published:** 2022-10-26

**Authors:** Di Wu, Xiao-Ling Wang, Xi-Xia Zhu, Hai-Hong Wang, Wei Liu, Lin Qi, Peng Song, Ming-Ming Zhang, Wei Zhao

**Affiliations:** 1College of Agronomy, Henan University of Science and Technology, Luoyang 471003, China; 2Anyang Yindu Agricultural and Rural Bureau, Anyang 455000, China

**Keywords:** ammonia-oxidizing bacteria, Italian ryegrass, regrowth, cytokinin, root–shoot signal

## Abstract

Potted Italian ryegrasses (*Lolium multiflorum* L.) were used to investigate the effect of ammonia-oxidizing bacterial (AOB) strain that coexisted in rhizosphere soil on Italian ryegrass regrowth. The results showed that the isolated and screened AOB strain (S2_8_1) had 100% similarity to *Ensifer sesbaniae*. The inoculation of S2_8_1 on day 44 before defoliation caused its copy number in rhizosphere soils to increase by 83–157% from day 34 before defoliation to day 14 after defoliation compared with that in Italian ryegrass without S2_8_1 inoculation, indicating that S2_8_1 coexisted permanently with Italian ryegrass. The coexistence promoted the delivery of root-derived cytokinin to leaves and to increase its cytokinin concentrations; thus, the Italian ryegrass regrowth accelerated. During the 14-day regrowth period, the S2_8_1 coexistence with Italian ryegrass caused its leaf and xylem sap cytokinin concentrations, rhizosphere soil nitrification rates, net photosynthetic rates, and total biomass to increase by 38%, 58%, 105%, 18%, and 39% on day 14 after defoliation, respectively. The inoculation of S2_8_1 on day 2 before defoliation also increased the regrowth of Italian ryegrass. Thus, the coexistence of AOB with Italian ryegrass increased its regrowth by regulating the delivery of cytokinins from roots to leaves.

## 1. Introduction

Animal husbandry, which is based on forage production, is very important for meeting people’s requirement for meat and milk [1]. Forage regrowth refers to its growth of stems and leaves after defoliation. Recently, for the purpose of improving their production, the regrowth of Kikuyu grass (*Pennisetum clandestinum* Hochst. Ex Chiov), perennial ryegrass (*Lolium perenne* L.), tall fescue (*Festuca arundinacea* Schreb.), Bermuda grass (*Cynodon dactylon* (L.) Pers.), and forage peanut (*Arachis pintoi*) have been studied greatly [2,3,4,5]. Inoculation with plant-growth-promoting bacteria has played an increasing role in forage grass growth [6,7,8]. Thus, the increase effect of microbes on forage grass regrowth is critical for protoculture development. 

Many plant organs such as leaves and roots are places where cytokinins are produced; however, their synthesis mainly occur in roots, where they are delivered to other organs to adjust plant growth and development [9,10]. Furthermore, cytokinins play essential roles in plant response to nitrate, where they act as secondary messengers in plants [11,12]. Root-derived cytokinins induced by soil nitrogen (NO_3_^−^) also play an important role in forage regrowth under the participation of growth resources. For example, forage grass regrowth involves photosynthate accumulation, and plant growth resources of water, carbon dioxide, and mineral nutrients are involved in photosynthesis. Nitrogen fertilizer plays an important role in ryegrass regrowth [13] and NO_3_^−^ could improve ryegrass’ regrowth by stimulating its roots to synthesize cytokinins, which are delivered to leaves to increase photosynthesis [14,15]. 

Ammonia-oxidizing bacteria (AOB) are the main soil microorganisms involved in nitrification, which plays a role on soil supplying NO_3_^−^ to plants [16]. The rhizosphere is a good habitat for the colonization of soil AOB. As nutrients are apt to accumulate in the rhizosphere micro-environment [17,18], AOB are usually found in fertile soils [19,20]. Wang et al. [21] reported that the AOB strain *Ensifer sesbaniae* colonized in ryegrass rhizosphere soil and promoted its regrowth. The rhizosphere is an area where many bacteria colonize and play a role in host plant growth [22]. It is a good place for soil AOB to coexist with forage grasses. However, reports on the relationship between soil AOB and forage grass regrowth are limited, and few studies have reported the abundance of soil AOB in forage grass rhizosphere. The present study explored the relationships among the abundance of rhizosphere soil AOB, soil nitrification rate, root-derived cytokinin, and regrowth of forage grass to reveal the regrowth mechanism from a coexistence point of view of soil microorganism and plant. Other studies have shown that some cytokinin-producing bacteria could promote the growth of wheat or maize [23,24]. The present study investigated the effect of root-derived cytokinin caused by soil AOB on crop growth.

In this study, Italian ryegrass was selected as the test materials for studying forage grass regrowth. Italian ryegrass grows rapidly, and it is tolerant to intensive defoliation and the most widely planted forage grass worldwide [25,26]. The present study aimed to reveal that a rhizosphere soil AOB strain could coexist with Italian ryegrass and improve its regrowth. To test this hypothesis, one strain of AOB was isolated and screened from the soil of the study site. Then, the isolated AOB strain was inoculated into the soil to detect its abundance by quantitative fluorescence polymerase chain reaction (PCR). Quantitative PCR is an accurate method for detecting the number of bacteria in soils [27]. The relationship between the abundance of the isolated AOB strain and soil nitrification rates in rhizosphere soils, leaf cytokinin and leaf photosynthesis were investigated to understand the regrowth mechanisms.

## 2. Materials and Methods

### 2.1. Experimental Design

#### 2.1.1. Isolation of Soil AOB Strain

Several types of media were prepared for the isolation and screening of soil AOB strains. The enrichment medium consisted of 0.5 g of (NH_4_)_2_SO_4_, 0.75 g of KH_2_PO_4_, 0.25 g of NaH_2_PO_4_, 0.01 g of MnSO_4_∙4H_2_O, 0.03 g of MgSO_4_∙7H_2_O, 5.0 g of CaCO_3_, and 1000 mL of distilled water (pH 7.2). The separation medium consisted of 1 L of enrichment medium and 2.5 g of agar, and the LB medium consisted of 10 g of tryptone, 5 g of yeast extract, 10 g of NaCl, and 1000 mL of distilled water (pH 7.0). 

Approximately 50 g of soil, which was sampled in the experimental farm of Henan University of Science and Technology, Luoyang city, Henan Province, China, was added to the enrichment media for culture for 15–25 days at 28 °C in the dark. Then, the enrichment media were examined with Griess reagent to qualitatively test the presence of the AOB strain. Griess reagent is a qualitative test reagent for nitrite, which could verify if ammonia was oxidized. The enrichment media (0.5 mL) were dropped in a white porcelain colorimetric plate, followed by the addition of 0.1 mL of Griess reagent to the white plate to color. The occurrence of red, pink, or dark red indicated nitrite ions existence, AOB strain existence, and ammonia oxidated into nitrite, respectively.

A portion of enrichment culture solution showing AOB strain growth was transferred to a new enrichment medium for continued culture. This process was repeated two times to obtain a pure AOB enrichment solution. Then, a portion of the solution was applied to separation media for culture for 5–8 days at 28 °C in the dark. Next, single tip-size bacterial colonies were selected and spread onto the separation medium of the agar plate. This process was repeated 5–10 times to obtain pure bacterial colonies. Finally, one strain (S2_8_1) of bacteria was selected as the experimental strains. The pure bacterial colonies were subsequently inoculated into the enrichment medium solution, which ultimately served as the AOB solutions used in this study. In addition, a portion of the AOB solution was inoculated into the LB medium to study the heterotrophic growth of the bacterial strain. The isolated AOB strains were examined through Gram staining.

#### 2.1.2. Nitrification Rate Detection

The purified AOB culture (1 mL) and its control consisting of 1 mL liquid medium without AOB were applied to 50 g soil samples to test the AOB effects on soil nitrification. The soil samples had an organic carbon content of 24.3 g·kg^−1^ and a total nitrogen content of 2.2 g·kg^−1^. Then, 20 g of each soil sample was cultivated for 7 days at 25 °C under a moisture level equal to 60% field capacity. The differences in soil NO_3_^−^ content between the inoculated soil samples and those not inoculated were divided by 7 (for the 7 days) to obtain the soil nitrification rate per day.

#### 2.1.3. Identification of AOB Strains

Pure bacterial colonies in the separation medium were sampled for AOB identification. A bacterial genome extraction kit (Bioer Technology Co., Ltd., Hangzhou, China) was used to extract the total bacterial DNA, and the 16S rDNA was amplified. The forward primer used was 27F (5′-AGAGTTGATCCTGGCTCAG-3′), and the reverse primer used was 1492R (5′-TACCTTGTTACGACTT-3′). Each 25-µL PCR reaction mixture included 2.5 µL of 10× buffer (with Mg^2+^), 2 µL of dNTP (2.5 mmol·L^−1^), 0.4 µL each of forward and reverse primers (10 µmol·L^−1^), 30 ng of DNA template, and 0.75 U of Ex Taq DNA polymerase (Bioer Technology Co., Ltd., Hangzhou, China). DdH_2_O was added until the final volume was reached. The reaction procedure was as follows: pre-degeneration at 94 °C for 4 min; degeneration at 94 °C for 1 min; and 30 cycles of renaturation at 55 °C for 45 s, extension at 72 °C for 2 min, and extension at 72 °C for 10 min. The amplification products were analyzed using 1.0% agarose gels. A DNA fragment purification kit was used to purify approximately 1.5-kb-long target fragments. The purified products were sequenced by TinyGene Bio-Tech (Shanghai) Co., Ltd. The DNA sequencing results were analyzed and compared with the sequence information in GenBank to identify closely related strains. The neighbor-joining method of MEGA 3.1 was used to construct phylogenetic trees of AOB. 

#### 2.1.4. Ryegrass Regrowth Following Defoliation 

##### Experimental Site and Soil Conditions

A pot experiment was performed at Henan University of Science and Technology, Luoyang City, Henan Province, China. The site receives an average annual rainfall of 601 mm and has an average temperature of 14.2 °C. An Italian ryegrass cv. Barwoltra (Barenbrug, Beijing, China) was selected as the experimental grass. In early March 2020, Barwoltra seeds were sown in pots placed in a greenhouse with a light transmittance of 90% and a temperature of 25 °C and allowed to grow for 2 weeks until they reached the seedling stage. The seedlings were watered every 2–3 days to maintain proper growth. 

Two hundred plastic pots with a diameter of 20 cm, height of 25 cm, and volume of 13.5 L were used. Each pot was filled with 5.8 kg of brown soil, which had a bulk density of 1.35 g·cm^−3^, an organic carbon content of 23.7 g·kg^−1^, a total nitrogen content of 2.2 g·kg^−1^, an available phosphorus content of 10.5 mg·kg^−1^, an available potassium content of 121.6 mg·kg^−1^, and an available magnesium content of 263.5 mg·kg^−1^ (pH 7.7). The soil organic carbon, total nitrogen, available phosphorus, available potassium, and available magnesium contents were determined by the potassium dichromate outside-heating method, Kjeldahl method, molybdenum antimony resistance colorimetric method, flame photometry, and atomic absorption spectrometry, respectively; the soil pH was measured via potentiometry.

##### Experimental Design

Uniform and vigorously growing seedlings were transplanted into 200 pots with six seedlings in each pot, and then the transplanted seedlings were placed in the sun and allowed to grow. On day 10 after transplanting, 48 pots with uniform and vigorously growing seedlings were randomly selected for the experiments. Figure 1 displays the treatment parameters, trial time course, and related indicators measured for the 48 pots. These 48 selected pots were divided into four groups of 12 pots each to investigate the effect of the AOB strain on Italian ryegrass regrowth following defoliation. These groups were subjected to one of four treatments: (1) regrowth with S2_8_1 strain inoculated relatively long before defoliation (TL), (2) regrowth with liquid medium without AOB added relatively long before defoliation (TA), (3) regrowth with S2_8_1 strain inoculated shortly before defoliation (TS), and (4) regrowth with liquid medium without AOB added shortly before defoliation (TB). The three pots in each subgroup were considered three replicates for each measurement of each treatment. The four subgroups in each treatment were randomly named subgroups 1, 2, 3, and 4.

From day 54 after transplanting to day 68 after transplanting, the 14-day regrowth period was selected. For inoculation of AOB, 200 mL of purified S2_8_1 culture (216,090 cfu·mL^−1^) was added to the soils of TS and TL treatments on days 52 and 10 after transplanting (on days 2 and 44 before defoliation), respectively. Meanwhile, 200 mL of liquid medium without AOB was added to the soils of TA and TB treatments on 52 and 10 days after transplanting (on days 2 and 44 before defoliation) as the control, respectively.

##### Measurement Process

On day 20 after transplanting (on day 34 before defoliation), the subgroup 1 in each treatment was picked out to sample the rhizosphere soils and measure Italian ryegrass biomass. These sampled rhizosphere soils were used for the measurements of AOB stain number and soil organic carbon content. 

At the beginning of the regrowth period, namely day 0 during the 14-day period, four steps were carried out in subgroup 2 of each treatment. First, their net photosynthetic rate (P_n_), transpiration rate (T_r_), and stomatal conductance (G_s_) were measured. Then, they were taken to the laboratory and clipped to a height of 5 cm. The clippings were weighed immediately to determine their fresh weight. Moreover, 0.5 g of the fresh clippings was sampled to measure the concentrations of zeatin riboside in the leaves. The remaining leaf samples were dried for 72 h at 65 °C to determine the dry matter content, which was calculated by dividing the dry weight by the fresh weight. The clipping biomass was obtained by multiplying the leaf dry matter content by the clipping fresh weight.

Second, the xylem sap was collected in accordance with the methods of Wang et al. [28]. After the Italian ryegrass leaf blades were clipped, the wounds were immediately covered with 1.0 g of absorbent cotton for 12 h to absorb the xylem sap. The xylem sap volume was determined by dividing the increase in cotton weight by 1 g·cm^−3^. The cotton was then compressed, and the sap was squeezed out. The collected xylem sap was used for measuring the zeatin riboside concentrations.

Third, the intact roots and soils were removed from the pots to sample the rhizosphere soils in subgroup 1 of each treatment. A clean brush was used to brush off the soil particles from all the roots. This process was repeated 5–10 times until tiny soil particles could be peeled off from the roots. The soil particles were collected separately during each brushing to avoid mixing with each other. The soil close to the roots was regarded as rhizosphere soil. The collected rhizosphere soil was used for the measurement of soil nitrification rate, soil NH_4_^+^ and NO_3_^−^ contents and AOB stain number. 

Fourth, the roots and shoot of the plants in subgroup 1 of each treatment were separated. The roots were washed with water to remove any soil particles. These roots and shoot were dried for 72 h at 65 °C to determine the dry matter content, which was used for calculating their biomass.

After the measurements of subgroup 1 of each treatment were completed, the plants in subgroups 3 and 4 were clipped to a 5 cm height and allowed to regrow for 7 days, and then the clippings were dried to obtain their biomass. 

On day 7 of the regrowth period, similar to subgroup 2 on day 0 during the regrowth period, the same four steps were carried out for subgroup 3. The plants in subgroup 4 were clipped to a 5 cm height and allowed to regrow for another 7 days, and the clippings were dried to measure their biomass. On day 14 of the regrowth period, the four steps were similarly carried out for subgroup 4. Thus, the plants in each treatment were clipped two times on days 0 and 7 during the 14-day period. The 7-day regrowth period was used because regrowth differences are likely to be discernible during this period.

#### 2.1.5. Measurements and Data Analysis

##### Biomass, Photosynthesis, Soil Nitrification, Zeatin Riboside 

According to Figure 2, subgroups 1 and 2 were used to determine the aboveground and total biomasses of each treatment on day 34 before defoliation and day 0 of the regrowth period, respectively; they were determined by the sums of the clipping and shoot biomasses and of the clipping, shoot, and root biomasses, respectively. The regrown leaf biomass of each treatment on day 7 after the first and second clippings referred to the clipping biomass in subgroups 3 and 4, respectively. On day 7 after the first clipping, the clipping and shoot biomass of subgroup 3 was added with its clipping biomass on day 0 to be considered the aboveground biomass of each treatment at this time. The aboveground biomass of subgroup 3 was added with its root biomass to be used to determine the total biomass of each treatment at this time. The same method was used to determine the aboveground and total biomasses of each treatment on day 7 after the second clipping.

An LI-6400 photosynthesis-measuring device was used at 11:00 am, and the P_n_, T_r_, and G_S_ were measured, with the light intensity, temperature, and carbon dioxide concentration set to 1000 μmol·m^−2^·s^−1^, 28 °C, and 400–420 ppm, respectively. Due to the narrow leaf blades of Italian ryegrass, a single leaf could not completely cover the leaf chamber of the photosynthesis equipment. Thus, 2–3 leaves partly overlapped tightly to enlarge the leaf area and cover the leaf chamber.

Soil NH_4_^+^ and NO_3_^−^ contents were measured using the indophenol blue method and phenol disulfonic acid colorimetry, respectively [29]. Soil samples whose moisture was maintained at 60% field capacity were cultured for 7 days at 25 °C to determine the soil net nitrification rate. The differences in soil NO_3_^−^ content before and after culturing were divided by 7 days to obtain the soil net nitrification rate of each day.

Xylem sap extract was collected by cutting the stem, followed immediately by covering the stem wound with 1.0 g of absorbent cotton. After 12 h, the cotton weight was determined. The quantity of xylem sap was equal to the increased weight of the absorbent cotton. The volume of sap was determined by dividing the increased weight by 1 g/cm^3^. Then, cotton was compressed at the end of a 10 mL syringe by using a piston. The sap from each sample was pooled into 5 mL centrifuge tubes to measure the concentrations of zeatin riboside in the xylem sap. After extraction, all of the xylem sap collected was immediately injected into a solid-phase extraction C-18 column (Waters Corporation, Milford, MA, USA) to filter, blow-dried with nitrogen. The leaf sample was extracted with 80% methanol containing 1 mmol/L di-tert-butyl-4-methylphenol. The leaf extract was also filtered with the solid-phase extraction C-18 column and blow-dried with nitrogen. The residues of the leaf and xylem sap extract samples were dissolved in 0.01 mol/L phosphate buffer solution (pH 7.4) and subjected to an enzyme-linked immunosorbent assay to determine zeatin riboside. The zeatin riboside concentrations in the leaf and xylem sap were estimated using the enzyme-linked immunosorbent assay as described by Qin and Wang [30]. The test kits for zeatin riboside were produced at the Phytohormone Research Institute of China Agricultural University. The amounts of zeatin riboside were divided by their leaf sample weight to determine their concentrations in leaves or divided by the volume of xylem sap to obtain the zeatin riboside concentrations in the xylem sap (C_ZR_). The transport rates of zeatin riboside from roots to leaves in the darkness (R_ZR_) were expressed as the amounts of zeatin riboside in the xylem sap per hour. Quantitative real-time PCR and analysis

The total DNA of the soil genome was extracted from rhizosphere soil samples by using the MO-BIO PowerSoil DNA Isolation Kit and used as a template for PCR amplification. The specific primers F (5′-ATGTACTGCGCTCAAATCCGA-3′), R (5′-ATGATGAAGGCAAAACCACGAT-3′), and probe P (5′-FAM-ACAACGCAGAAGTCGCACGGAAG-BHQ1-3′) of S2_8_1 were used for the PCR amplification of genomic DNA targeting the gene of S2_8_1, which was obtained by 16Sr DNA amplification in “Section 2.1.3. Identification of AOB strains”. The 25 µL reaction system contained the following: 12.5 µL Premix Ex Taq (qPCR probe, 2×), 0.5 µL forward primer F (10 µM), 0.5 µL reverse primer R (10 µM), 0.5 µL probe (10 µM), 5 µL DNA template, and 6 µL ddH_2_O. The reaction conditions were predenaturation at 95 °C for 30 s, denaturation at 95 °C for 10 s, annealing at 60 °C for 45 s, and recycling for 45 times. Each soil sample was tested thrice. The extracted total DNA of the soil genome was amplified, and the Ct value of the sample obtained by quantitative fluorescence PCR was introduced into the standard curve equation to calculate the copy number of S2_8_1 (copies/g) in the rhizosphere soil of Italian ryegrass. The copy number was used to indicate the bacterial number.

The specific primers of S2_8_1 were amplified using PCR to construct the standard curve. The amplified products were purified, and the plasmids were constructed and transformed into *Escherichia coli* competent cells. The *Escherichia coli* containing the target gene plasmid in the clone library was cultured in a shake flask at 37 °C. The plasmid was extracted by Axygen Plasmid Miniprep Kit (Axygen). The plasmid concentration was determined by Qubit 3.0 (Life Biotech) and the copy number of the plasmid was calculated. The gradient dilution results of the standard plasmid were as follows (5–7 points were generally diluted, and the points with good qPCR were selected as the standard curves): 5.86 × 10^5^, 5.86 × 10^4^, 5.86 × 10^3^, 5.86 × 10^2^, and 5.86 copies/µL. Five standard samples with different concentrations were subjected to quantitative fluorescence detection to establish a linear relationship between the Ct value and the concentration of S2_8_1 strain in Italian ryegrass rhizosphere soil. The abbreviations used in the text are summarized in Table 1.

All values given in the figure are average values. The general linear model in SPSS 23 was used to conduct one-way analysis of variance followed by Dunnett test at the 0.05 probability level.

## 3. Results

### 3.1. Screening and Identification of AOB Strains

The AOB strain S2_8_1 was isolated and screened in this study. As shown in Figure 3, it belongs to *Ensifer* and has 100% similarity to *Ensifer sesbaniae*. Colonies appeared at 4–6 and 1–2 days after the inoculation of S2_8_1 into the separation and LB media, respectively (Figure 4A,B). The growth potential of S2_8_1 was higher in the LB medium than in the separation medium, indicating that S2_8_1 had mixed nutritional characteristics and was inclined to be heterotrophic. On the AOB separation medium, the colonies of S2_8_1 appeared to be milky white and round with an irregular edge and smooth surface (Figure 4A). The S2_8_1 bacterium was 1.09 μm long and 0.54 μm wide, and it had a shape similar to a short bar under the microscope (Figure 4C). S2_8_1 greatly increased the soil nitrification rate. The soil nitrification rates were 10.89 and 5.89 mg·kg^−1^·d^−1^ in soils with and without S2_8_1 added, respectively, and 1.85 times higher in soil with S2_8_1 added than in soil without S2_8_1 added.

### 3.2. Biomass 

Figure 5 shows that on day 34 before defoliation the aboveground and total biomasses were significantly higher in TL than in TA, TB, and TS. On days 7 and 14 after defoliation, the regrown leaf, aboveground and total biomasses were significantly higher in TS than in TB and in TL than in TA. These results indicate that S2_8_1 inoculation greatly increased ryegrass growth or regrowth. TS treatment had significantly higher regrown leaf biomass than TL on days 7 and 14 after defoliation, suggesting that S2_8_1 inoculation shortly before defoliation had a larger promotional effect on the accumulation of biomass than the inoculation relatively long before defoliation. The regrowth also increased biomass accumulation, given that similar aboveground and total biomasses were between TS and TL treatments on day 0 after defoliation; however, after 14-day regrowth, a significant increase in these biomasses was observed in TS compared with TL on day 14 after defoliation.

### 3.3. Photosynthesis

On days 0, 7, and 14 after defoliation, P_n_ was significantly higher in TS than in TB and in TL than in TA. On day 7 after defoliation, G_s_ and T_r_ were significantly higher in TS than in TB and in TL than in TA (Figure 6). These results showed that S2_8_1 inoculation promoted photosynthesis of ryegrass before or after defoliation. P_n_ increased significantly in TS than in TL on days 7 and 14 after defoliation, indicating that S2_8_1 inoculation shortly before defoliation had a larger promotional effect on photosynthesis than the inoculation relatively long before defoliation.

### 3.4. Soil Nitrification Rate, Soil Ammonium and Nitrate Nitrogen

As shown in Figure 7, significant increases in rhizosphere soil nitrification rates were observed in the TS compared with TB and in TL compared with TA on days 0, 7, and 14 after defoliation. Therefore, S2_8_1 inoculation promoted rhizosphere soil nitrification before or after defoliation. Significantly higher soil nitrification rates in the rhizosphere environments were detected in TS than in TL on days 7 and 14 after defoliation, indicating that S2_8_1 inoculation shortly before defoliation had a larger promotional effect on rhizosphere soil nitrification than the inoculation relatively long before defoliation. During the regrowth period, irregular results; that is, by chance, were observed in rhizosphere soil’s NO_3_^−^ and NH_4_^+^ contents. Thus, the inoculation of S2_8_1 could not increase their contents.

### 3.5. Plant Zeatin Riboside

On days 0, 7, and 14 after defoliation, the leaf zeatin riboside content, R_ZR_, and C_ZR_ of TS and TL significantly increased compared with those of TB and TA, respectively (Figure 8). Thus, S2_8_1 inoculation promoted leaf cytokinin contents and delivery rates from the roots to the leaves. Zeatin riboside is one of the major cytokinin forms. The leaf zeatin riboside content, C_ZR_, and R_ZR_ were significantly higher in TL than in TS on days 7 and 14 after defoliation. Therefore, the inoculation of S2_8_1 shortly before defoliation had a larger promotional effect on leaf cytokinin content and its delivery rates from the roots to the leaves than the inoculation relatively long before defoliation.

### 3.6. Strain Number and Soil Organic Carbon Content in Rhizosphere Soil

The copy number of S2_8_1 in rhizosphere soil was significantly higher in TL than in TA, TB, and TS on day 34 before defoliation, and significantly higher in TS than in TB and in TL than in TA on days 0, 7, and 14 after defoliation, suggesting that the inoculation of S2_8_1 increased its number in rhizosphere soil. Although the copy number of S2_8_1 was higher in TS than in TL on days 0, 7, and 14 after defoliation, no significant difference was found between them. Thus, the copy number of S2_8_1 tended to increase in soil with S2_8_1 inoculation shortly before defoliation compared with its inoculation relatively long before defoliation. Moreover, no significant differences were observed for rhizosphere soil’s organic carbon contents among all treatments through the trial period (Figure 9).

## 4. Discussion

### 4.1. Coexistence of S2_8_1 and Italian Ryegrass 

In this study, a stable coexisting relationship was found between Italian ryegrass and S2_8_1 in TL. The copy number of S2_8_1 increased averagely twice in TL compared with TA from day 34 before defoliation to the end of the trial. This finding showed that the inoculation of S2_8_1 in TL on day 44 before defoliation caused some of it to colonize in rhizosphere soil, and the S2_8_1 colonization in rhizosphere soil continued to exist until the regrowth period. In addition, S2_8_1 increased growth and regrowth of Italian ryegrass, given that the total biomass before defoliation and during regrowth period were approximately 1.46 and 1.39 times higher in TL than TA, respectively. 

Soil AOB comprise autotrophic, heterotrophic, and mixotrophic AOB [31,32,33]. In the present study, the AOB strain S2_8_1, which belongs to *Ensifer*, has mixotrophic characteristics, but is inclined to be heterotrophic. Preece et al. [34] and Gargallo-Garriga et al. [35] used organic substances as the substrate for growing heterotrophic AOB. Carbon is the chief element of organic compounds, and soil organic carbon may represent the number of organic compounds in soil [36]. The rhizosphere soil organic carbon contents in TL slightly changed compared with that in TA through the trail period, indicating that rhizosphere micro-environment sustained the stable soil organic carbon in TL. This finding was the main reason that after the colonization of S2_8_1, its coexistence with Italian ryegrass could last until regrowth period in the rhizosphere micro-environment. 

Given its S2_8_1 content being 216,090 cfu·mL^−1^, the 200 mL inoculation solution had a large number of S2_8_1, which caused more of it to easily enter into the rhizosphere micro-environment to colonize when inoculating. This finding was the main reason for the high copy number of S2_8_1 tending to occur in the rhizosphere soil of TS compared with TL. However, no statistically significant differences were observed in the copy number of S2_8_1 between TL and TS during the regrowth period. The explanation for this finding was that similar rhizosphere soil organic carbon contents in TL and TS caused a similar S2_8_1 copy number of rhizosphere soil. These findings further showed that rhizosphere soil organic matter was the key factor promoting S2_8_1 to colonize in the rhizosphere micro-environment. 

The S2_8_1 in TL increased Italian ryegrass growth and regrowth because on days 0, 7, and 14 after defoliation, the total biomass of TL increased by 43%, 38%, and 39% compared with that of TA, respectively. The S2_8_1 of TS likewise promoted the growth and regrowth of Italian ryegrass, given that 43–58% increases in total biomass occurred in TS compared with TB from day 0 to day 10 after defoliation.

The total biomass on day 14 after defoliation increased by 12% in TS compared with that in TL, showing that S2_8_1 had large promoting effect on Italian ryegrass regrowth in TS compared with TL. This finding may be ascribed to the high copy number of S2_8_1 in TS. In addition, the interval time between inoculation and defoliation was a potential influencing factor. Ye et al. [37] and Wang et al. [38] reported that the effect of bacteria strain inoculation faded over time. After S2_8_1 inoculation, only 2 days were left before defoliation in TS. However, the S2_8_1 inoculation and Italian ryegrass defoliation in TL had an interval of 54 days.

### 4.2. Root–Derived Cytokinins

The results revealed that the coexistence of S2_8_1 with Italian ryegrass in TL continuously increased the amount of root cytokinin delivered to leaves, thus improving its regrowth. 

Considering AOB are the main bacteria involved in soil nitrification, S2_8_1 caused the rhizosphere soil nitrification rates to increase by 1.48 times in TL compared with that TA during regrowth. A high rhizosphere soil nitrification rate caused the continuous release of more NO_3_^−^ from the soil during the rewatering period, increasing the stimulation by NO_3_^−^ in the roots. Soil NO_3_^−^ could stimulate roots to synthesize cytokinins in various plants [12,39]. As a result, more root cytokinins were synthesized and continuously delivered to increase the leaf cytokinin content in TL than in TA. The cytokinins synthesized in the roots could be transmitted to the leaves by means of xylem sap [40,41]. R_ZR_ indicates the delivery speed of cytokinins from roots to leaves in the darkness. The R_ZR_ increasing by 51% in TL compared with that in TA during the regrowth period demonstrated the additional cytokinins being synthesized and delivered to the leaves in darkness. The cytokinin delivery rate to the leaves in the presence of sunlight was the product of T_r_ and C_ZR_, and transpiration drove the sap to transmit from roots to leaves under this condition. More cytokinins were synthesized and delivered to the leaves in TL than in TA in sunlight during the regrowth period. High C_ZR_ and T_r_ were present in TL compared with TA during the regrowth period. In addition, during the regrowth period, C_ZR_ and T_r_ increased by 49% and 8% in TL compared with those in TA, respectively.

Similarly, high leaf cytokinin content occurred in TS compared with TB or TL, because the S2_8_1 in TS increased the rhizosphere soil nitrification rates and cytokinin delivery from roots to leaves. As mentioned above, the high copy number of S2_8_1 and the short interval time between inoculation and defoliation contributed to high rhizosphere soil nitrification rates in TS. 

The leaf cytokinin content in TL and TS increased by 49% and 68%, respectively, compared with that in TA and TB during rewatering, which was beneficial for the increase in their leaf P_n_. Cytokinin plays a promotional role in photosynthesis [42,43]. The occurrence of high P_n_ in TL and TS than in TA and TB promoted their regrowth. Similarly, TS had high P_n_ compared with TL during the regrowth period. 

Wang et al. [14] observed that the root-originated cytokinins induced by the continued addition of NO_3_^−^ increased Italian ryegrass regrowth due to its short duration in soil. However, the present study revealed that the coexistence of S2_8_1 with Italian ryegrass increased its regrowth by inducing root cytokinin delivery to the leaves continuously. Many studies qualitatively explored the coexistence of rhizobium with leguminous crop or other microorganisms [44,45]. Different from these studies, the present study detected the amount of S2_8_1 in *Rhizobiaceae* by using quantitative PCR, revealing the coexistence of S2_8_1 and Italian ryegrass, and to explore the direct connections among S2_8_1, the nitrification, root–derived cytokinins, and regrowth of Italian ryegrass. The low increase effect of S2_8_1 on Italian ryegrass regrowth was observed in TL compared with TS. However, S2_8_1 coexistence with Italian ryegrass in TS showed significance for the feasibility of the technique when using soil heterotrophic AOB strains, which were isolated from soil similar to the present study in grassland production. For example, the inoculation of other soil heterotrophic AOBs after emergence could cause their coexistence with forage grasses, which could increase the regrowth of biomasses with each clipping, and decrease the production costs. On the contrary, timely inoculations before each clipping decreased production efficiency. Based on the present study, some companies of microbial biotechnology may produce new heterotrophic AOB strains to be used on grass production, and some new bacterium fertilizers involving heterotrophic AOB strains could be possibly produced.

## 5. Conclusions

The inoculation of S2_8_1 caused it to coexist with Italian ryegrass in the rhizosphere micro-environment. Such a coexistence increased the soil nitrification rates, which played an important role on cytokinin synthesis in the roots and its delivery to the leaves. As a result, the cytokinin concentration in the leaves increased. The increase in cytokinin improved the photosynthetic rate of leaves and was beneficial to the accumulation of photosynthetic organic matter, thereby promoting the regrowth of Italian ryegrass. Thus, the coexistence of AOB strain with Italian ryegrass increased its regrowth by regulating the delivery of root–derived cytokinins to leaves. These findings are beneficial for increasing forage grass production.

## Figures and Tables

**Figure 1 microorganisms-10-02122-f001:**
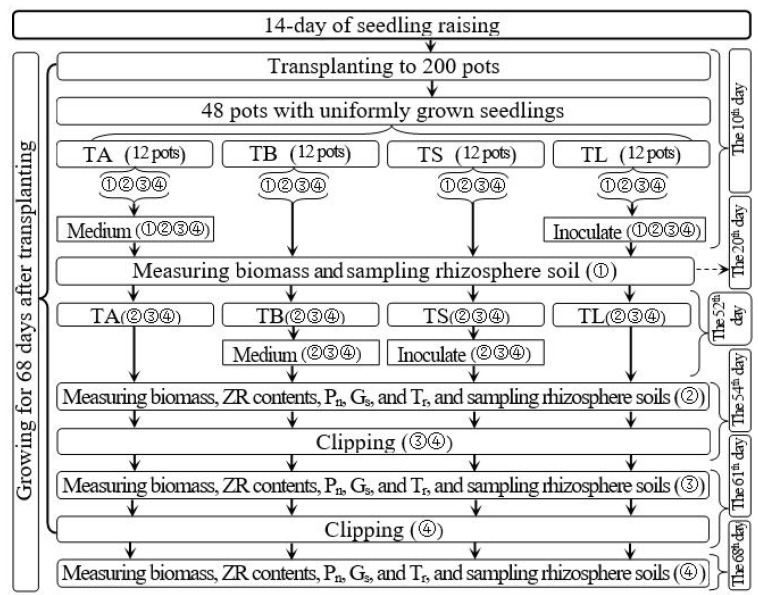
Schematic diagram for the experimental design. TA, TB, TS, TL indicate treatments of regrowth with liquid medium without AOB added relatively long before defoliation, regrowth with liquid medium without AOB added shortly before defoliation, regrowth with S2_8_1 strain inoculated relatively long before defoliation, and regrowth with S2_8_1 strain inoculated shortly before defoliation, respectively. “①”, “②”, “③”and “④” show the first, second, third and fourth subgroups of each treatment, respectively. The 10th, 20th, 52nd, 54th, 61st and 68th days refer to the 10th, 20th, 52nd, 54th, 61st and 68th of the 68 days after transplanting. ZR indicates zeatin riboside. P_n_, G_s_, and T_r_ represent the net photosynthetic rate, stomatal conductance, and transpiration rate, respectively.

**Figure 2 microorganisms-10-02122-f002:**
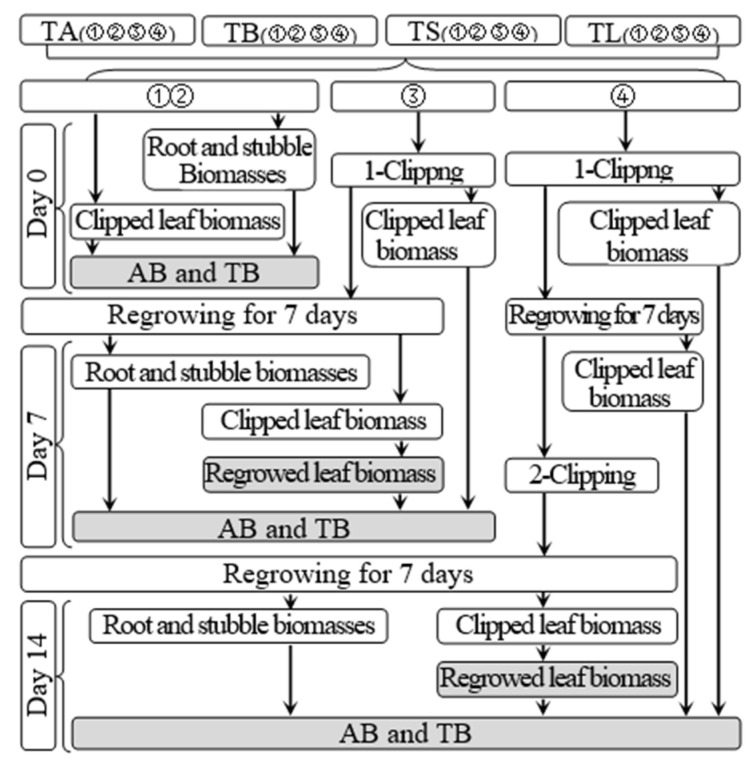
Schematic diagram for the measurement of biomass. TA, TB, TS, TL indicate treatments of regrowth with liquid medium without AOB added relatively long before defoliation, regrowth with liquid medium without AOB added shortly before defoliation, regrowth with S2_8_1 strain inoculated relatively long before defoliation, and regrowth with S2_8_1 strain inoculated shortly before defoliation, respectively. “①”, “②”, “③”and “④” show the first, second, third and fourth subgroups of each treatment, respectively. “day 0”, “day 7”, and “day 14”, respectively stand for the beginning, 7th day, 14th days of the 14-day regrowth period. “AB” and “TB” refer to accumulated aboveground and total biomasses, respectively.

**Figure 3 microorganisms-10-02122-f003:**
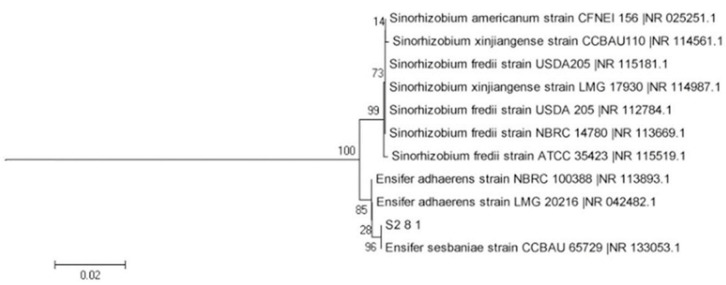
Phylogenetic tree based on the 16S rDNA sequence of 1 AOB strains. The S2_8_1 in the figure is the strains that has been isolated and screened in the present study.

**Figure 4 microorganisms-10-02122-f004:**
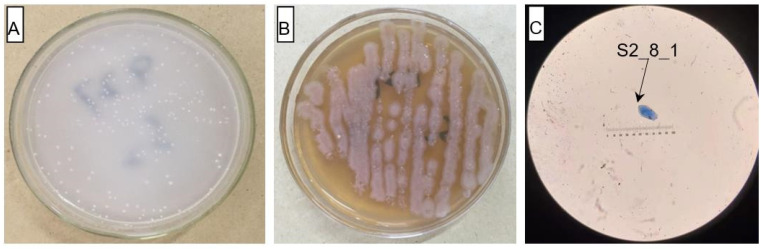
Cell and colonial morphology of strain S2_8_1 in the study. The S2_8_1 was the strain that has been isolated and screened in the present study. (**A**) and (**B**) Showed the morphology characteristics of cell and colony of strain S2_8_1 in separation and LB mediums, respectively. (**C**) Showed the size and sharp of strain S2_8_1, respectively.

**Figure 5 microorganisms-10-02122-f005:**
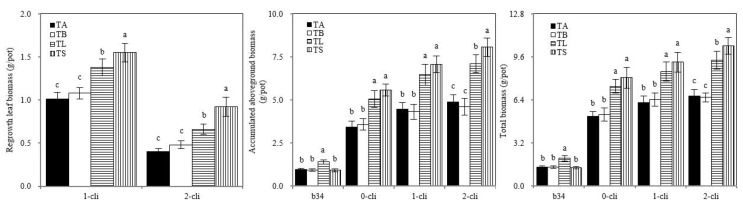
Biomass of Italian ryegrass in the different treatments in the experiment. TA, TB, TS, TL indicate treatments of regrowth with liquid medium without AOB added relatively long before defoliation, regrowth with liquid medium without AOB added shortly before defoliation, regrowth with S2_8_1 strain inoculated relatively long before defoliation, and regrowth with S2_8_1 strain inoculated shortly before defoliation, respectively. “b34”, “0-cli”, “1-cli” and “2-cli” mean, respectively, day 34 before defoliation, days 0, 7 and 14 after defoliation. Biomass refers to dry matter. The values are the mean ± standard error (n = 3). The different letters in each row indicate significant differences (*p* < 0.05).

**Figure 6 microorganisms-10-02122-f006:**
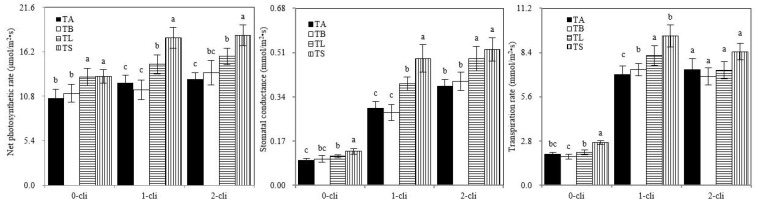
Net photosynthetic rate, transpiration rate, and stomatal conductance of Italian ryegrass in the different treatments in the experiment. TA, TB, TS, TL indicate treatments of regrowth with liquid medium without AOB added relatively long before defoliation, regrowth with liquid medium without AOB added shortly before defoliation, regrowth with S2_8_1 strain inoculated relatively long before defoliation, and regrowth with S2_8_1 strain inoculated shortly before defoliation, respectively. “0-cli”, “1-cli” and “2-cli” mean, respectively, days 0, 7 and 14 after defoliation. The values are the mean ± standard error (n = 3). The different letters in each row indicate significant differences (*p* < 0.05).

**Figure 7 microorganisms-10-02122-f007:**
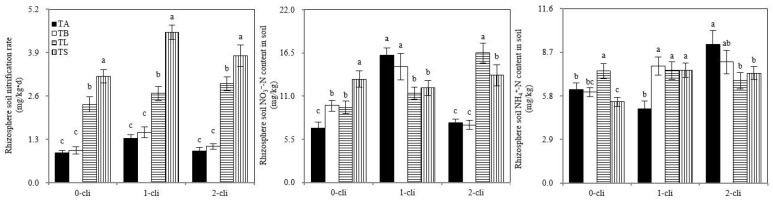
Rhizosphere soil nitrification rate, rhizosphere soil ammonium and nitrate nitrogen contents of Italian ryegrass in the different treatments in the experiment. TA, TB, TS, TL indicate treatments of regrowth with liquid medium without AOB added relatively long before defoliation, regrowth with liquid medium without AOB added shortly before defoliation, regrowth with S2_8_1 strain inoculated relatively long before defoliation, and regrowth with S2_8_1 strain inoculated shortly before defoliation, respectively. “0-cli”, “1-cli” and “2-cli” mean, respectively, days 0, 7 and 14 after defoliation. The values are the mean ± standard error (n = 3). The different letters in each row indicate significant differences (*p* < 0.05).

**Figure 8 microorganisms-10-02122-f008:**
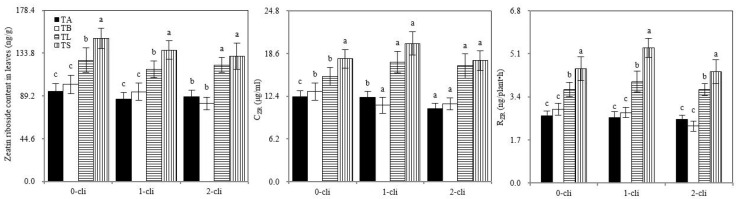
Zeatin riboside concentrations in the newly grown leaves, R_ZR_ and C_ZR_ in the xylem saps in different treatments in the experiment. TA, TB, TS, TL indicate treatments of regrowth with liquid medium without AOB added relatively long before defoliation, regrowth with liquid medium without AOB added shortly before defoliation, regrowth with S2_8_1 strain inoculated relatively long before defoliation, and regrowth with S2_8_1 strain inoculated shortly before defoliation, respectively. “0-cli”, “1-cli” and “2-cli” mean, respectively, days 0, 7 and 14 after defoliation. C_ZR_ mean zeatin riboside concentrations in xylem sap. R_ZR_ mean delivery rates of zeatin riboside from roots to leaves in the darkness. The values are the mean ± standard error (n = 3). The different letters in each row indicate significant differences (*p* < 0.05).

**Figure 9 microorganisms-10-02122-f009:**
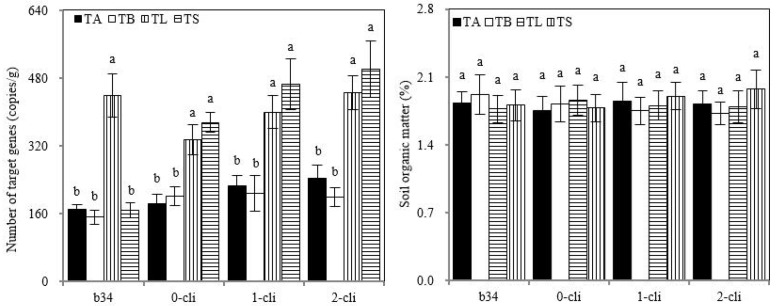
S2_8_1 number in rhizosphere soil and soil organic matter in the study. TA, TB, TS, TL indicate treatments of regrowth with liquid medium without AOB added relatively long before defoliation, regrowth with liquid medium without AOB added shortly before defoliation, regrowth with S2_8_1 strain inoculated relatively long before defoliation, and regrowth with S2_8_1 strain inoculated shortly before defoliation, respectively. “b34”, “0-cli”, “1-cli” and “2-cli” mean, respectively, day 34 before defoliation, days 0, 7 and 14 after defoliation. The values are the mean ± standard error (n = 3). The different letters in each row indicate significant differences (*p* < 0.05).

**Table 1 microorganisms-10-02122-t001:** Symbol definition.

Symbol	Definition	Symbol	Definition
TA	regrowth with liquid medium without AOB added shortly before defoliation	S2_8_1	The AOB strains
TS	regrowth with S2_8_1 strain inoculated shortly before defoliation	R_ZR_	Delivery rate of zeatin riboside from roots to leaves
TB	regrowth with liquid medium without AOB added relatively long before defoliation	C_ZR_	zeatin riboside concentration in xylem sap
TL	regrowth with S2_8_1 strain inoculated relatively long before defoliation	P_n_	Photosynthetic rate
NO_3_^−^	Soil nitrate	G_s_	Stomatal conductance
NH_4_^+^	Soil ammonium	T_r_	Transpiration rate
AOB	Ammonia oxidizing bacteria	PCR	Polymerase chain reaction

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
