# Peer review of "Effect of Ammonia-Oxidizing Bacterial Strains That Coexist in Rhizosphere Soil on Italian Ryegrass Regrowth"

_microorganisms, 2022, doi:10.3390/microorganisms10112122_

Round 1
Reviewer 1 Report
This seems to be well-conducted research and to have clear, repeatable results. The scientific content contributes to the microbiology research field. The work was provided with a sufficient level of scientific novelty.
The introduction part is well described. rework it to connect all parts of the introduction. The authors described the philosophy of the proposed research in the study of the relationship between the abundance of rhizosphere soil ammonia-oxidizing bacteria and forage grass regrowth will reveal its mechanism from a coexisting view.
The hypothesis part has some limits. In my opinion, is necessary to analyze better the cytokine regulation in plants. Include the missing information (research gaps and the significance of your research compared with other crops).
The introduction provides a good understanding of the subject and its importance, with a significant quantity of information. I suggest adding more theoretical information about the metabolism and role of cytokinins in plants. Root–shoot signalling of the cytokinins section of the paper is quite confusing. Authors use too many abbreviations, and the reader is lost after first lines. I recommend rewriting it.
The theoretical and practical reasons for this paper for the experiments are very reasonable.
The Paper is very complex and innovative. The structure of the paper is logical, and the results are well reproduced.
The analytical work of the authors is perfect. I have no critical comments. The results reported have not been published elsewhere.
My questions/suggestions:
The material and methods part is understandable. How many measurements of photosynthesis did the authors take?
I suggest that authors pay more attention to terminology, units, and measured values.
Line 229 - incorrect units of light intensity (1000 lx) - change to μmol.m-2s-1.
In Fig 6 they are incorrect units for photosynthesis, stomatal conductance (Pn - μmol m−2 s−1; Gc, T - mmol m−2 s−1)
The correct abbreviation of stomatal conductance is not Sc, but Gc.
Graphs have to be self-readable. Do not use abbreviations in legends.
I didn't find in the text meaning of the abbreviation ZR.
I recommend improving the description of the method of cytokinin analysis.
I think the overall concept is interesting and potentially important. I recommend to ACCEPT the paper for publication with major revision.
Author Response
Overall comments
This seems to be well-conducted research and to have clear, repeatable results. The scientific content contributes to the microbiology research field. The work was provided with a sufficient level of scientific novelty.
The introduction part is well described. rework it to connect all parts of the introduction. The authors described the philosophy of the proposed research in the study of the relationship between the abundance of rhizosphere soil ammonia-oxidizing bacteria and forage grass regrowth will reveal its mechanism from a coexisting view.
The hypothesis part has some limits. In my opinion, is necessary to analyze better the cytokine regulation in plants. Include the missing information (research gaps and the significance of your research compared with other crops).
The introduction provides a good understanding of the subject and its importance, with a significant quantity of information. I suggest adding more theoretical information about the metabolism and role of cytokinins in plants. Root–shoot signalling of the cytokinins section of the paper is quite confusing. Authors use too many abbreviations, and the reader is lost after first lines. I recommend rewriting it.
The theoretical and practical reasons for this paper for the experiments are very reasonable.
The Paper is very complex and innovative. The structure of the paper is logical, and the results are well reproduced.
The analytical work of the authors is perfect. I have no critical comments. The results reported have not been published elsewhere.
Response:
Thank you very much for your overall positive evaluation of our manuscript. We are very glad that the Reviewer highly evaluated our manuscript, and provided constructive comments and valuable suggestions that have helped us to further improve the quality of our manuscript. We have addressed all of your queries and improved our manuscript following your suggestions as you can see in our point-to-point responses to your comments below.
Comment 1
The hypothesis part has some limits. In my opinion, is necessary to analyze better the cytokine regulation in plants. Include the missing information (research gaps and the significance of your research compared with other crops).
Response:
Thank you very much for this nice suggestion. For the purpose of improving hypothesis, some references have been added to describe “the cytokine regulation in plants” (Line 41-52); and some other references also have been added to state “research gaps and the significance of your research compared with other crops” (Line 69-70).
Comment 2
The introduction provides a good understanding of the subject and its importance, with a significant quantity of information. I suggest adding more theoretical information about the metabolism and role of cytokinins in plants. Root–shoot signalling of the cytokinins section of the paper is quite confusing. Authors use too many abbreviations, and the reader is lost after first lines. I recommend rewriting it.
Response:
We are thankful to the Reviewer for this constructive comment. According to your suggestion, new references that indicate the metabolism and role of cytokinins have been added. (Line 41-52, 69-70)
The abbreviation “ZR” has been replaced by “zeatin riboside”. (Line 213, 415, 416, 419, 420, 426, 432; Figure 8)
The abbreviation “SOC” has been replaced by “soil organic carbon”. (Line 444, 453, 474, 478, 487)
“Root–shoot signalling of the cytokinins” has been replaced by “root-derived cytokinin” in the manuscript. (Line 17, 45, 67, 506, 553, 574)
Comment 3
The material and methods part is understandable. How many measurements of photosynthesis did the authors take?
Response:
Thank you very much for this query. As shown in Figure 1, we measured photosynthesis three times. And the three measurements for the photosynthesis also showed in “2.1.4.3. Measurement process”. (Line 193)
The first measurement of photosynthesis in the Line 199-200.
The second measurement of photosynthesis in the Line 229-230.
The third measurement of photosynthesis in the Line 232-233.
Comment 4
I suggest that authors pay more attention to terminology, units, and measured values.
Response:
Reviewer has rightly raised the question. Following your suggestion, the terminology has been modified. (Line 33-35, 315)
The unit has been modified. (Line 187-188, 190, 252, 481)
The measured values are correct.
Comment 5
Line 229 - incorrect units of light intensity (1000 lx) - change to μmol.m-2s-1.
Response:
Thank you very much for pointing this out. Following your suggestion, “lx” has been modified to “μmol·m−2·s−1”. (Line 252)
Comment 6
In Fig 6 they are incorrect units for photosynthesis, stomatal conductance (Pn - μmol m−2 s−1; Gc, T - mmol m−2 s−1).
Response:
Thank you very much for pointing out this important issue. In Figure 6, “Net photosynthetic rate (Pn)” and “Transpiration rate (Tr)” units are correct.
The “Stomatal conductance (Gs)” unit “mol/m2·s” has been changed to “mmol/m2·s”.
Comment 7
The correct abbreviation of stomatal conductance is not Sc, but Gc.
Response:
Thank you very much for pointing out this important issue. According to your suggestion, “Sc” has been replaced with “Gs”. (Figure 1; Table 1; Line 184, 200, 251, 379)
Comment 8
Graphs have to be self-readable. Do not use abbreviations in legends.
Response:
We are thankful to the Reviewer for this constructive comment. According to your suggestion, it has been modified.
Due to the drawing requirements, the abbreviation in Figure 1 cann’t be changed.
The “Pn” has been modified to “Net photosynthetic rate”, the “Tr” has been modified to “Transpiration rate”, the “Gs” has been modified to “Stomatal conductance” in Figure 6.
The “ZR” has been modified to “zeatin riboside” in Figure 8.
Comment 9
I didn't find in the text meaning of the abbreviation ZR.
Response:
Thank you very much for this nice suggestion. The abbreviation “ZR” has been replaced by “zeatin riboside”. (Line 213, 415, 416, 419, 420, 426, 432; Figure 8)
Comment 10
I recommend improving the description of the method of cytokinin analysis.
Response
We appreciate the reviewer for pointing out this shortage. Following your suggestion, new analysis method have been added. (Line 271-296)

Reviewer 2 Report
In general, the authors have done a good job explaining the background information necessary to appreciate the rationale and results of the experiments. The manuscript was prepared correctly. Methodology and analysis of results rather don't raise any objections.
However, some minor amendments are needed. The discussion of the results is written a bit generally. There are papers related to this topic that the authors did not cite. An assessment putting the findings into perspective and make a solid conclusion is missing. The authors should emphasize more the novelty and usefulness of the results.
Author Response
Overall comments
In general, the authors have done a good job explaining the background information necessary to appreciate the rationale and results of the experiments. The manuscript was prepared correctly. Methodology and analysis of results rather don't raise any objections.
However, some minor amendments are needed. The discussion of the results is written a bit generally. There are papers related to this topic that the authors did not cite. An assessment putting the findings into perspective and make a solid conclusion is missing. The authors should emphasize more the novelty and usefulness of the results.
Response:
Thank you very much for your overall positive evaluation of our manuscript. We are very glad that the Reviewer highly evaluated our manuscript, and provided constructive comments and valuable suggestions that have helped us to further improve the quality of our manuscript. We have addressed all of your queries and improved our manuscript following your suggestions as you can see in our point-to-point responses to your comments below.
Comment 1
However, some minor amendments are needed. The discussion of the results is written a bit generally. There are papers related to this topic that the authors did not cite. An assessment putting the findings into perspective and make a solid conclusion is missing. The authors should emphasize more the novelty and usefulness of the results.
Response:
Thank you very much for this nice suggestion. Following your suggestion, specific results have been added to reduce “The discussion of the results is written a bit generally”. (Line 458-459, 466-467, 497-498, 511-512, 521-522, 528-530, 536-538)
Two references have been added to discuss the topic. (Line 549-550)
An assessment has been added to put the findings into perspective, to make a solid conclusion and to emphasize the novelty. (Line 550-554)
It has been added to emphasize the usefulness of the results. (Line 562-565)

Round 2
Reviewer 1 Report
The authors revised the manuscript according to the comments thoroughly and respond to the comments point by point, at present, the manuscript could be accepted.